# Sexual health needs of female sex workers in Côte d'Ivoire: a mixed-methods study to prepare the future implementation of pre-exposure prophylaxis (PrEP) for HIV prevention

Valentine Becquet [1,2] Marcellin Nouaman,[3] Mélanie Plazy,[4] Jean-Marie Masumbuko,[3] Camille Anoma,[5] Soh Kouame,[6] Christine Danel,[3] Serge Paul Eholie,[3] Joseph Larmarange,[2] for the ANRS 12361 PrEP-CI Study group

For numbered affiliations see end of article.

**Correspondence to**
Dr Valentine Becquet;
valentine.becquet@ined.fr

## ABSTRACT

**Objective** To describe sexual and reproductive health (SRH) needs of female sex workers (FSWs) to inform the future implementation of pre-exposure prophylaxis (PrEP) for HIV prevention in this population.

**Design and setting** The ANRS 12361 PrEP-CI cross-sectional and mixed-methods study was designed and implemented with two community-based organisations in Côte d'Ivoire.

**Participants** A convenience sample of 1000 FSWs aged ≥18, not known as HIV-positive, completed a standardised questionnaire assessing sociodemographic characteristics, sexual practices, use of community health services and a priori acceptability of PrEP. Twenty-two indepth interviews and eight focus group discussions were also conducted to document FSWs' risky practices and sexual behaviours, experiences with violence and discrimination, attitudes regarding HIV and sexually transmitted infections (STIs), and barriers to SRH services.

**Results** Although 87% described consistent condom use with clients, more than 22% declared accepting condomless sexual intercourse for a large sum of money. Furthermore, condom use with their steady partner and knowledge of their partner's HIV status were low despite their acknowledged concurrent sexual partnerships. While inconsistent condom use exposed FSWs to STIs and undesired pregnancies, the prevalence of contraceptive strategies other than condoms was low (39%) due to fear of contraception causing sterility. FSWs faced obstacles to accessing SRH care and preferred advice from their peers or self-medication.

**Conclusions** Despite adoption of preventive behaviour in most cases, FSWs are still highly exposed to HIV. Furthermore, FSWs seem to face several barriers to accessing SRH. Implementing PrEP among FSWs in West Africa, such as in Côte d'Ivoire, constitutes an opportunity to consider the regular follow-up of HIV-negative FSWs. PrEP initiation should not condition access to SRH services; conversely, SRH services could be a way to attract FSWs into HIV prevention. Our

### Strengths and limitations of this study

► This study combines quantitative survey to document sexual and health behaviours and needs of female sex workers (FSWs) and qualitative survey to understand the rationale behind these behaviours and needs.

► This study used a convenience sample representative of FSWs actually reached by two community non-governmental organisations, but is not representative of the overall population of FSWs.

► This is a comprehensive descriptive study but not powered enough for multivariate explicative analysis.

results highlight the importance of developing a people-focused approach that integrates all SRH needs when transitioning from PrEP efficacy trials to implementation.

## INTRODUCTION

Despite global progress in reducing new HIV infections and AIDS-related deaths in the last 10 years in sub-Saharan Africa,[1 2] current policies and programmes are focusing on the identification of HIV-infected people in order to link them to HIV care and treatment,[3 4] knowing that antiretroviral treatment has been proven to reduce HIV transmission.[5] However, the number of new HIV infections remains too high to achieve epidemic control.[6] Numerous trials have shown that oral pre-exposure prophylaxis (PrEP), when taken correctly, was highly efficacious to prevent HIV acquisition, in particular among men having sex with men (MSM).[7–9] Since 2015, oral PrEP has been recommended by the WHO for populations at 'substantial risk' of HIV acquisition.[10]

In West Africa, most countries have mixed HIV epidemics, with a relatively low prevalence in the general population (compared with that in Eastern and Southern Africa) but severely affected key populations, particularly female sex workers (FSWs) and MSM.[11] In Côte d'Ivoire in 2012, the HIV prevalence was estimated to be 29% among FSWs[12] and 19% among MSM.[12] The National Program Against HIV/AIDS (Programme National de la Lutte contre le Sida, PNLS) requested operational research and data on the relevance of PrEP in order to better consider its future integration in the national algorithm of HIV prevention. A PrEP demonstration project (ANRS 12324-EF CohMSM) is currently being implemented among MSM. Our research team was invited to explore the sexual health needs of FSWs in this country as a potential target for a future PrEP programme, knowing that PrEP was not yet available.

Although PrEP is effective when taken properly, the FEM-PrEP[13] and VOICE (Vaginal and Oral Interventions to Control the Epidemic)[14] trials conducted among women from the general population in Southern and Eastern Africa showed low adherence to the treatment, resulting in a low or even null effect of PrEP. Similarly, PrEP implementation trials conducted among FSWs in Africa showed varying results regarding retention. In Benin, the retention rate after 10 months was 66%,[15] and the overall retention rate after a complete follow-up of 28 months was 48%.[16] In South Africa, it was 22% after 12 months[17] despite a high declared acceptability of PrEP before the implementation.[18] Moreover, PrEP constitutes a new HIV prevention tool but does not prevent sexually transmitted infections (STIs) or unwanted pregnancies. It is therefore necessary to consider the overall needs of target populations in terms of sexual and reproductive health (SRH).[19] More operational and social science research is needed for the implementation of PrEP to be a success, especially on creating demand for oral PrEP, improving adherence, understanding the social and behavioural impact of PrEP, and integrating PrEP services with other services.[10]

In this context, in order to design a future PrEP programme targeting FSWs, the ANRS 12361 PrEP-CI pilot study was implemented to explore sexual healthcare needs that should be considered within such a programme and to better describe FSWs currently reached by peer educators. This paper aims to describe the work and social environment of FSWs, their SRH needs, and possible barriers to accessing care in two different settings in Côte d'Ivoire, that is, different elements that need to be taken into account when implementing PrEP. We adopted a mixed approach: a quantitative survey was used to reach a high number of FSWs in order to be able to calculate the incidence of HIV infection and to compare sexual and health behaviours and needs of FSWs in the two settings; qualitative interviews were conducted to understand the rationale behind these behaviours and needs.

## METHODS
### Study setting
The ANRS 12361 PrEP-CI cross-sectional and mixed-methods study was designed and implemented with two Ivorian community-based organisations between September 2016 and March 2017. Aprosam works within the city of San Pedro and in the surrounding areas, particularly in villages close to farming businesses (coffee and cocoa exploitation). Espace Confiance operates in several districts of Abidjan, the economic capital of Côte d'Ivoire (Koumassi, Marcory, Treichville, Zone 4 and Port-Bouët with its beaches). Both of these non-governmental organisations (NGOs) deliver HIV prevention and testing services directly at prostitution sites (outreach activities) and provide HIV and SRH care services for MSM and FSWs through a community clinic. Recruitment of participants for this study was made possible by the Aprosam and Espace Confiance organisations' networks of peer educators and their access to the population. The purpose of the quantitative study was not to be representative of all FSWs in Côte d'Ivoire but rather to represent FSWs who could be reached by the two partner NGOs and who could potentially benefit from PrEP in a future programme.

### Quantitative analysis of a survey questionnaire
From October 2016 to January 2017, a convenience sample of 1000 FSWs were recruited either by peer educators or when FSWs visited NGOs' community clinics. Eligibility criteria for the quantitative survey included being 18 years or older, working at a prostitution site at the time of the survey, and being HIV negative or of unknown HIV status at the time of the survey. Peer educators enrolled FSWs who met the eligibility criteria and agreed to participate after reading an information sheet and signing a consent form.

FSWs answered a face-to-face, 45-item standardised paper questionnaire that assessed their sociodemographic characteristics (age, nationality, level of education, number of children), their sexual practices and behaviours (duration and location of sex work, usual price of sexual intercourse, condom use with clients and regular partners, assault/coerced sexual intercourse), their knowledge and use of community health services (medical consultations, hepatitis B immunisation, declared STIs, sex work during menstruation, use and knowledge of contraception, undesired pregnancies, abortion), a priori acceptability of a PrEP offer (perception of the risk to contract HIV, knowledge of any medicine (traditional or modern) to prevent HIV infection, interest in a modern medicine for HIV prevention, acceptance of a medical follow-up every 3 months) and HIV monitoring (regularity of HIV testing, knowledge of HIV status of regular partners). Data collection was carried out by peer educators in dedicated health centres and prostitution sites.

FSWs were also tested for HIV, and in case of a positive result dried blood spot sampling was performed to

determine the window of infection through a recent infection testing algorithm adapted to the Ivorian context and thus describe HIV incidence in this population. Individuals diagnosed with HIV during the study were referred to the community clinics by peer educators for HIV care and treatment.

We described the sociodemographic characteristics, sexual behaviours and reproductive health of participants surveyed according to the study setting (Abidjan and San Pedro). Due to the fact that it is not a randomly taken sample but rather a convenience sample of women reached by the two NGOs, statistical tests such as Pearson's $\chi^2$ test or Fisher's exact test could not be formally used to compare the two study settings. Missing data were excluded from percentage calculations. All analyses were performed with Stata V.12.0 software.

## Qualitative analysis of interviews

In addition to the quantitative survey, a qualitative study was conducted from November 2016 to December 2016 among a convenience sample of 66 FSWs recruited during the outreach activities of peer educators and with the aim to reach a maximum of different profiles of women (in terms of age, number of years working as FSWs, type of prostitution site). On each prostitution site visited, we decided to perform indepth interviews or focus group discussions (FGDs), depending on the practicality of the site (ambient noise, opportunity for privacy) and the time allowed. Every time, according to the type of interviews to be performed, we conducted one to three individual interviews and/or one focus group with five to eight FSWs.

Data were collected at prostitution sites in and around Abidjan and San Pedro by a female researcher in demography (first author), who carried out 22 indepth interviews (duration: 30–60 min) and 8 FGDs (duration: 60–150 min) using a semistructured interview guide. FSWs were interviewed about their sociodemographic characteristics (age, nationality, level of education, number of children, number of dependents, partner/husband), sex work (entry into prostitution, duration and location of sex work, usual price of sexual intercourse, mobility, regular clients, work during menstruation, future perspectives), risky practices and sexual behaviours (condom use depending on the type of practices, main perceived risk of unprotected sex, current possession of condoms, negotiation of (un)protected sex with clients and regular partners), community dynamics (relationship with the pimp, the owner of the prostitution site, other FSWs, peer educators from NGOs, source of help in case of money, health or administrative issues), experiences with violence and discrimination (physical/moral violence from clients, partners, authorities, experiences of stigmatisation, barriers to accessing healthcare or administrative procedures), knowledge and attitudes regarding HIV and STIs (perception of the risk to contract HIV depending on the type of practices, perception of the global risk to contract HIV and means used for

prevention, frequency and location of HIV testing, physical signs of STIs and means used for treatment), barriers to healthcare (untreated health issues, locations of care seeking), use of drugs and alcohol, barriers to and need for SRH services (knowledge and screening of cervical cancer, hepatitis B and C, tuberculosis, knowledge and use of contraception including emergency contraception, knowledge and recourse to social workers), and a priori acceptability of a PrEP offer (knowledge of any medicine (traditional or modern) to prevent HIV infection, interest in a modern medicine for HIV prevention, acceptance of a medical follow-up every 3 months, issues arisen after the presentation of PrEP).

Each of the indepth interviews and FGDs was recorded (except for two participants who refused as they were afraid to be recognised), transcribed and uploaded into NVivo software (QSR International, V.11 Pro, 2016) by the qualitative interviewer. She also conducted the qualitative analysis following two principles. First, a cross-sectional review, based on questions derived from the discussion guide, allowed for a thematic analysis. Data collected provide great information on FSWs' sexual and health behaviours and needs, preferences and social trajectories. However, many themes are not included in this paper and will be addressed in a further paper. The main themes explored for this article were related to access to care (visit of community health centres, use of mobile clinics, referral by peer educators, barriers to access to care and stigmatisation) and to the potential interest, utility of PrEP and obstacles (unprotected sexual intercourse with clients and partners, risk perception and women's priorities, mobility and working periods, access to condoms, use of contraception, anticipation of high-risk sex and violent clients). We remained open to new themes as they emerged from the data in an inductive manner (eg, condom breakage and self-medication). Second, we reviewed each interview or FGD as a whole in order to identify the chain of events leading each woman to not access healthcare or to not use condoms, for example. Quotes presented here were translated verbatim from French to English by the authors.

## Patient and public involvement

No patient was involved in the research design nor in the conduct of the study. Peer educators of six different community NGOs were involved in the development of the research questions during a workshop. The two selected NGOs in Abidjan and San Pedro participated in the design, recruitment and conduct of the study. Data from the quantitative survey and qualitative interviews were disseminated among the community through peer educators, who helped in the interpretation of results.

## RESULTS
### Main characteristics of participants
The characteristics of FSWs who participated in the quantitative survey are presented in table 1. The median age

**Table 1** Main characteristics of participants in the quantitative survey

| Variables | All women, N=1000 n (%) | San Pedro, n=400 n (%) | Abidjan, n=600 n (%) |
|---|---|---|---|
| Median age (IQR) (years) | 25 (21–29) | 25 (22–30) | 24 (21–28) |
| Age (years) | | | |
| ≤24 | 470 (47.0) | 168 (42.0) | 302 (50.3) |
| 25–34 | 431 (43.1) | 181 (45.2) | 250 (41.7) |
| ≥35 | 99 (9.9) | 51 (12.8) | 48 (8.0) |
| Level of education | | | |
| No school | 220 (22.1) | 115 (28.9) | 105 (17.6) |
| Primary school | 382 (38.4) | 163 (40.9) | 219 (36.7) |
| Secondary school/university | 393 (39.5) | 120 (30.2) | 273 (45.7) |
| Missing | 5 | 2 | 3 |
| Nationality | | | |
| Ivorian | 690 (69.0) | 312 (78.0) | 378 (63.0) |
| Foreign | 310 (31.0) | 88 (22.0) | 222 (37.0) |
| Has a boyfriend/husband | | | |
| Yes | 714 (71.9) | 317 (80.7) | 397 (66.2) |
| No | 279 (28.1) | 76 (19.3) | 203 (33.8) |
| Missing | 7 | 7 | 0 |
| Number of children | | | |
| 0 | 426 (43.1) | 132 (33.3) | 294 (49.7) |
| 1 | 301 (30.5) | 122 (30.8) | 179 (30.2) |
| 2 | 155 (15.7) | 82 (20.7) | 73 (12.3) |
| ≥3 | 106 (10.7) | 60 (15.2) | 46 (7.8) |
| Missing | 12 | 4 | 8 |
| Frequency of sex work | | | |
| Every day or almost every day | 743 (75.3) | 275 (69.3) | 468 (79.5) |
| Sometimes | 243 (24.7) | 122 (30.7) | 121 (20.5) |
| Missing | 14 | 3 | 11 |
| How many years sex work has been practised | | | |
| ≤2 | 479 (47.9) | 176 (44.0) | 303 (50.5) |
| ≥3 | 521 (52.1) | 224 (56.0) | 297 (49.5) |
| Practised sex work in more than one city | | | |
| Yes | 268 (26.9) | 198 (49.7) | 70 (11.7) |
| No | 727 (73.1) | 200 (50.3) | 527 (88.3) |
| Missing | 5 | 2 | 3 |
| Where/how clients are contacted* | | | |
| Brothel | 302 (30.2) | 114 (28.5) | 188 (31.3) |
| Beach | 129 (12.9) | 71 (17.7) | 58 (9.7) |
| Bar/'maquis' | 471 (47.1) | 200 (50.0) | 271 (45.2) |
| Street | 145 (14.5) | 47 (11.8) | 98 (16.3) |
| By phone (through hotel owners) | 216 (21.6) | 123 (30.7) | 93 (15.5) |
| Hotel | 265 (26.5) | 156 (39.0) | 109 (18.2) |
| Home | 131 (13.1) | 66 (16.5) | 65 (10.8) |
| Number of clients during last day of work | | | |

Continued

**Table 1** Continued

| Variables | All women, N=1000 n (%) | San Pedro, n=400 n (%) | Abidjan, n=600 n (%) |
|---|---|---|---|
| ≤4 | 706 (70.8) | 233 (58.3) | 473 (79.2) |
| ≥5 | 291 (29.2) | 167 (41.7) | 124 (20.8) |
| Missing | 3 | 0 | 3 |
| How much did the last client pay (in Francs CFA, US$) | | | |
| ≤1999 (~3.50) | 238 (23.8) | 152 (38.0) | 86 (14.3) |
| 2000–4999 (3.50–8.75) | 287 (28.7) | 138 (34.5) | 149 (24.8) |
| 5000–9999 (8.75–17.50) | 241 (24.1) | 69 (17.2) | 172 (28.7) |
| ≥10 000 (17.50) | 234 (23.4) | 41 (10.3) | 193 (32.2) |
| Ever suffered assault/coerced sexual intercourse | | | |
| Yes | 115 (11.7) | 41 (10.5) | 74 (12.6) |
| No | 866 (88.3) | 351 (89.5) | 515 (87.4) |
| Missing | 19 | 8 | 11 |

*Most female sex workers meet their clients in more than one location: the total is not equal to 100%.

was 25 (IQR=22–30) years in San Pedro and 24 (IQR=21–28) years in Abidjan. Compared with FSWs reached in Abidjan, those reached in San Pedro were less educated, more often Ivorian, more likely to be the mother of at least one child, were paid less money and worked less regularly but much more frequently in more than one city. FSWs in San Pedro were also more often in a relationship, and the interviews showed that their boyfriend was often their pimp.

In the qualitative study, out of 66 interviewed FSWs, 26 agreed to provide their age. The median age was 28 (IQR=22–33) years. It became evident during the interview process that three FSWs were underaged (<18). The interview participants were mostly Ivorian (n=44); the remaining third (n=22) was Nigerian.

### High HIV exposure despite the use of condoms

Overall, in the questionnaire, 79% of FSWs in San Pedro and 92% of FSWs in Abidjan reported consistent condom use with their clients (table 2). However, the question about regular use of condoms could not fully capture actual condom use; there were several situations where FSWs had unprotected sexual intercourse. Twenty-three per cent would accept condomless sex for a large sum of money. This exposure to unprotected sex was reported as well by several FSWs during the qualitative interviews and was explained by the critical need for money.

> And when you look back at your week, you didn't even make 2000 francs. You begin to think about it. Ah! Honestly, I do accept [unprotected sex]. (FGD, San Pedro)

Several interviewed women also attested that violent clients had assaulted them and refused to use condoms.

> They brutalize us. Often, they don't wear any condom. They force us. Often even, young junkies, they can come upon us. And they assault us. (Indepth interview, San Pedro, 28 years old)

Moreover, 94% of FSWs in San Pedro and 89% of FSWs in Abidjan reported not systematically using condoms with their regular partner, even though only 11% and 23%, respectively, knew their partner's HIV status. This practice was reported during interviews as well, even though the women explained that they perceived a risk associated with condomless intercourse. During an FGD that took place in a slum in San Pedro, above a bar where FSWs meet their clients, interviewed women discussed about their regular partners. One of them stated that her boyfriend asked her to not use any condoms to prove her trust.

> This guy, he tells you I'm faithful to you. I want us to have sex without condoms to show trust. That's why I think that the scary person is your boyfriend, not the client. (FGD, San Pedro)

Another one explained the lack of trust in her partner was balanced by the fact that they would protect her from violent clients.

> Love is the only weapon where you sleep with your enemy (laughs). I mean, he's your closest enemy. He's the one who can kill you because he's not with you only. But you say, he's my official. You need him because he protects you. (FGD, San Pedro)

Even if the majority of FSWs declared regular condom use, most of them were still exposed to HIV: 59% had at least one instance of condomless intercourse over the previous week. FSWs' responses to the first question

**Table 2**  Condom use and HIV exposure in the quantitative survey

| Variables | All women, N=1000<br>n (%) | San Pedro, n=400<br>n (%) | Abidjan,<br>n=600<br>n (%) |
|---|---|---|---|
| Condom use with clients | | | |
| Never | 9 (0.9) | 3 (0.8) | 6 (1.0) |
| Sometimes | 29 (3.0) | 17 (4.4) | 12 (2.1) |
| Often | 86 (8.9) | 58 (15.1) | 28 (4.8) |
| Always | 837 (87.0) | 304 (79.2) | 533 (92.1) |
| Does not know/does not want to answer | 2 (0.1) | 2 (0.5) | 0 (0.0) |
| Missing | 37 | 16 | 21 |
| Use of condom with boyfriend/husband† | | | |
| Never | 370 (53.2) | 157 (50.5) | 213 (55.3) |
| Sometimes | 115 (16.5) | 61 (19.6) | 54 (14.0) |
| Often | 123 (17.7) | 49 (15.8) | 74 (19.2) |
| Always | 62 (8.9) | 19 (6.1) | 43 (11.2) |
| Does not know/does not want to answer | 26 (3.8) | 25 (8.0) | 1 (0.3) |
| Missing | 18 | 6 | 12 |
| Acceptance of condomless sexual intercourse in exchange for a large sum of money | | | |
| Never | 764 (77.4) | 251 (63.9) | 513 (86.4) |
| Sometimes | 79 (8.0) | 52 (13.2) | 27 (4.5) |
| Often | 92 (9.3) | 55 (14.0) | 37 (6.2) |
| Always | 12 (1.2) | 6 (1.5) | 6 (1.0) |
| Does not know/does not want to answer | 40 (4.0) | 29 (7.4) | 11 (1.9) |
| Missing | 13 | 7 | 6 |
| At least one instance of condomless intercourse over the last 7 days* | | | |
| Yes | 220 (58.8) | 152 (72.0) | 68 (41.5) |
| No | 154 (41.1) | 59 (28.0) | 95 (57.9) |
| Does not want to answer | 1 (0.2) | 0 (0.0) | 1 (0.6) |
| Missing | 625 | 189 | 436 |
| Last HIV test (months) | | | |
| <6 | 458 (45.9) | 230 (50.9) | 255 (42.6) |
| 6–12 | 239 (24.0) | 98 (24.6) | 141 (23.6) |
| ≥12 | 182 (18.2) | 69 (17.3) | 113 (18.9) |
| Never | 114 (11.4) | 26 (6.5) | 88 (14.7) |
| Does not know/does not want to answer | 4 (0.4) | 3 (0.7) | 1 (0.2) |
| Missing | 3 | 1 | 2 |
| Knowledge of boyfriend's/husband's HIV status† | | | |
| Yes | 121 (17.4) | 33 (10.6) | 88 (23.0) |
| No | 573 (82.3) | 279 (89.1) | 294 (76.7) |
| Does not want to answer | 2 (0.3) | 1 (0.3) | 1 (0.3) |
| Missing | 18 | 4 | 14 |
| Interest in a medicine protecting against HIV | | | |
| Yes | 982 (98.6) | 394 (99.0) | 588 (98.3) |
| No | 11 (1.1) | 2 (0.5) | 9 (1.5) |
| Does not know | 3 (0.3) | 2 (0.5) | 1 (0.2) |
| Missing | 4 | 2 | 2 |

Continued

**Table 2** Continued

| Variables | All women, N=1000 n (%) | San Pedro, n=400 n (%) | Abidjan, n=600 n (%) |
|---|---|---|---|
| If yes, would agree to a medical follow-up every 3 months | | | |
| Yes | 964 (99.4) | 391 (99.7) | 573 (99.1) |
| No | 5 (0.5) | 1 (0.3) | 4 (0.7) |
| Does not know | 1 (0.1) | 0 (0.0) | 1 (0.2) |
| Missing | 12 | 2 | 10 |

*This variable was added during the survey; for this reason, some participants did not answer the question.
†These variables do not concern all 1000 surveyed women but the 714 FSWs who declared having a boyfriend/husband.

assessing condom use might refer to 'typical use' as opposed to specific circumstances.

Regarding HIV testing, 51% of FSWs in San Pedro and 43% of FSWs in Abidjan had received their last HIV test less than 6 months ago. However, 7% in San Pedro and 15% in Abidjan had never been tested before the survey.

During each of the qualitative interviews and FGDs, we presented PrEP as a medicine that could protect against HIV if properly taken and explained that it would require regular medical follow-up. PrEP was not available in Côte d'Ivoire at the time of the interviews and women had never heard about it. However, several questions emerged in relation to concrete matters, such as side effects, cost, current availability in pharmacies, compatibility with pregnancy, appropriate reaction if one or more pills are forgotten, respect of the timing of daily administration, and so on. Many interviewed women considered PrEP as useful to prevent HIV transmission from their regular partners in particular, as they felt obligated to not use any condoms with them.

Danger itself, it comes from the one beside me. That pill is welcome, because by taking it I protect myself against the one beside me. (FGD, San Pedro)

PrEP was presented similarly, although more briefly, in the questionnaire. The large majority of surveyed FSWs (98.6%) showed interest in a medicine that could provide efficient protection against HIV. Of the FSWs interested in PrEP, 99.4% would agree to a medical follow-up every 3 months.

### Beyond HIV, many unmet SRH needs exist

In total, 43% of the survey participants reported at least one unwanted pregnancy and 50% had at least one abortion in their lifetime (table 3). Only 39% of surveyed FSWs were using a contraceptive method other than condoms; among them, most FSWs in Abidjan mentioned taking the pill (70%) compared with only 33% of FSWs in San Pedro, where 35% declared using an implant. Unfortunately, child desire was not asked in the quantitative survey, which does not allow us to calculate the unmet need for contraception among FSWs. However, as

a proxy, it appeared that most of the interviewed FSWs did not want a child at the moment.

My main risk, it is to not get pregnant because I'm still a schoolgirl. If I get pregnant, who will take care of it [the baby]? (Indepth interview, Abidjan, 18 years old)

However, women explained during interviews that they feared becoming sterile due to contraceptives, especially the pill.

They [peer educators from the community-based NGO] told me about the pill, but I refused because I don't have children yet. I don't want to have problems in the future. (Indepth interview, Abidjan, 18 years old)

That's what makes me tired. I'm afraid because I don't have children yet. That's my problem, otherwise for diseases, well, there are condoms. (Indepth interview, San Pedro, 19 years old)

Moreover, only half of the survey participants knew about emergency contraception, among whom 36% knew only non-medical means. Additionally, 36% of survey participants practised sex work during menstruation, mainly using tampons (62%) or cold water (24%) to stop the bleeding.

Finally, 79% of FSWs in San Pedro and 55% of FSWs in Abidjan reported contracting an STI over the past 12 months. Even though half of the questionnaire survey participants thought they were very exposed to HIV infection, the interviewed FSWs often declared being preoccupied by other diseases as well, such as STIs or cancer.

But we, every time, when we go in the bush, it's not only AIDS that kills. There are several diseases. Today we talk about cervical cancer. So I think it's not only AIDS we should get protected from. We have to protect ourselves from many diseases that are sexual. (FGD, San Pedro)

Despite the work of peer educators at prostitution sites, few surveyed FSWs visited the dedicated community clinics, with 76% in San Pedro and 61% in Abidjan

**Table 3** Sexual and reproductive health in the quantitative survey

| Variables | All women, N=1000 n (%) | San Pedro, n=400 n (%) | Abidjan, n=600 n (%) |
|---|---|---|---|
| Had at least one undesired pregnancy | | | |
| Yes | 416 (42.9) | 173 (45.1) | 243 (41.5) |
| No | 554 (57.1) | 211 (54.9) | 343 (58.5) |
| Missing | 30 | 16 | 14 |
| Had at least one abortion | | | |
| Yes | 488 (50.2) | 195 (50.4) | 293 (50.0) |
| No | 485 (49.8) | 192 (49.6) | 293 (50.0) |
| Missing | 27 | 13 | 14 |
| Use of contraception other than condom | | | |
| Yes | 391 (39.1) | 193 (48.3) | 198 (33.0) |
| No | 608 (60.8) | 206 (51.5) | 402 (67.0) |
| Does not know | 1 (0.1) | 1 (0.2) | 0 (0.0) |
| If yes, which contraceptive method* | | | |
| Pill | 204 (52.0) | 65 (33.5) | 139 (70.2) |
| Injectable | 91 (23.3) | 55 (28.5) | 36 (18.2) |
| Implant | 83 (21.2) | 68 (35.1) | 15 (7.6) |
| Other methods† (nivaquine, spice, traditional medicine, etc) | 24 (6.1) | 13 (6.7) | 11 (5.6) |
| Knowledge of emergency contraception | | | |
| Yes | 472 (48.4) | 195 (50.8) | 277 (46.9) |
| No | 497 (51.0) | 183 (47.7) | 314 (53.1) |
| Does not know | 6 (0.6) | 6 (1.5) | 0 (0.0) |
| Missing | 25 | 16 | 9 |
| If yes, type of emergency contraception known* | | | |
| Morning-after pill | 304 (64.1) | 100 (49.7) | 204 (74.7) |
| Other† (antibiotic, coffee, soda, salted water and lemon) | 170 (35.9) | 101 (50.3) | 69 (25.3) |
| Sex work during menstruation | | | |
| Yes | 363 (36.4) | 154 (38.5) | 209 (34.9) |
| No | 635 (63.6) | 246 (61.5) | 389 (65.1) |
| Missing | 2 | 0 | 2 |
| If yes, tool used for sex work during menstruation* | | | |
| Wash with ice-cold water | 86 (24.0) | 37 (24.2) | 49 (23.7) |
| Piece of ice | 26 (7.3) | 11 (7.2) | 15 (7.3) |
| Tampon | 222 (62.0) | 105 (68.6) | 117 (57.1) |
| Other tools† (hot water, soapy water, cotton, etc) | 49 (13.7) | 12 (7.8) | 37 (17.9) |
| Self-reported STI (last 12 months) | | | |
| Yes | 639 (64.7) | 312 (78.8) | 327 (55.2) |
| No | 349 (35.3) | 84 (21.2) | 265 (44.8) |
| Missing | 12 | 4 | 8 |
| Last medical consultation | | | |
| Less than 3 months | 195 (19.6) | 101 (25.4) | 94 (15.7) |
| 3–12 months | 475 (47.7) | 202 (50.8) | 273 (45.6) |
| More than a year | 258 (25.9) | 79 (19.8) | 179 (29.9) |
| Never consulted | 68 (6.8) | 16 (4.0) | 52 (8.7) |
| Missing | 4 | 2 | 2 |

Continued

**Table 3** Continued

| Variables | All women, N=1000 n (%) | San Pedro, n=400 n (%) | Abidjan, n=600 n (%) |
|---|---|---|---|
| If ever consulted, site of the last consultation with a doctor/nurse | | | |
| Dedicated facility | 225 (26.3) | 141 (40.5) | 84 (16.6) |
| Public facility | 415 (48.6) | 123 (35.3) | 292 (57.7) |
| Private facility | 213 (24.9) | 83 (23.8) | 130 (25.7) |
| Does not know | 1 (0.1) | 1 (0.3) | 0 (0.0) |
| Missing | 74 | 34 | 40 |

*Several possible answers.
†'Other' categories describe participants' specific answers.
STI, sexually transmitted infection.

consulting a health practitioner over the past year, among which 40% in San Pedro and 17% in Abidjan went to a dedicated facility. In interviews, some FSWs reported the inconvenient opening times and/or location, the fear of being identified as a FSW in the clinic area, and the stigmatising and judgemental attitudes of health professionals as reasons for not visiting these clinics. In the event of condom breakage, FSWs usually relied on self-medication. For example, a young woman described the beverages she would use for vaginal douche.

> I'm going to buy Coke with Nescafe. It's for cleaning everything falling down. (Indepth interview, Abidjan, 17 years old)

> They act similarly in case of suspicion of an STI.

> Before going to the clinic, we try traditional plants and medicines first. (FGD, San Pedro)

## DISCUSSION

Both the quantitative and qualitative results showed that FSWs were highly exposed to HIV despite their use of condoms. There were in fact a variety of situations in which the surveyed FSWs had condomless sex. First, the large majority did not use condoms with their regular partner despite their acknowledged concurrent sexual partnerships. Some women experienced coercion on the part of their male partners, questioning their faith in the relationship; having condomless sex was a proof of trust that was difficult to negotiate. Others used condomless sex as a negotiation strategy to obtain protection from their partners against the threat of violence. It seemed to be a calculated risk-mitigation strategy, although women were then exposed to the risk of HIV transmission. Second, some FSWs accepted condomless sexual intercourse for a large sum of money, especially when they had had few previous clients. Financial need associated with low prices of sexual intercourse and irregular weekly earnings drove some FSWs to engage in condomless sex as a way to earn more. In a context where gender norms reinforce male domination over women,[20] they consciously took risks when facing the primacy of men's sexual pleasure.[21] Third, the violence or the threat thereof that FSWs faced sometimes prevented them from negotiating condom use. Different studies showed that women who are victims of abuse are less likely to use condoms with their clients than those who are not.[22 23] Performing an illegal activity can also compel women, especially in the street, to negotiate quickly with clients at the expense of condom use. The situation appeared even worse in some rural areas around San Pedro, where the interviewed FSWs revealed that they could not buy any condoms in the village, as there was no point of sale.

Despite the adoption of preventive behaviour (condom use) in most cases, FSWs are still highly exposed to HIV, due to other unprotected sexual acts. In a complementary study within the PrEP-CI project (not yet published), we estimated the incidence among 1000 surveyed FSWs using a recent infection testing algorithm adapted to the Ivorian context. We found an incidence of 2.2 per 100 person-years (1.5 in Abidjan and 3.2 in San Pedro).[24] In such contexts, oral PrEP could be an appropriate and complementary preventive tool to cover situations where condom use cannot be negotiated.

The majority of interviewed and surveyed FSWs had low awareness and knowledge of PrEP before our study, but most of them were highly willing to use this medicine despite the constraints of regular medical follow-up. Participants felt PrEP would give added protection against infection, in particular with regular partners. A study in Kenya had similar findings and suggested to promote PrEP through outreach activities for sex workers.[25] However, a PrEP implementation trial in South Africa showed low adherence despite high declared acceptability before the implementation.[18] For this reason we were attentive to challenges that might hinder PrEP uptake and adherence for FSWs, such as side effects or timing of daily administration.

Our results showed that FSWs faced many unmet needs regarding SRH beyond HIV prevention and treatment. Inconsistent condom use exposed FSWs to STIs[26 27] and undesired pregnancies,[28] which could increase mortality

and morbidity.[29] The prevalence of contraceptive use was low in the surveyed population despite the high risk of undesired pregnancy due to the common fear of contraception causing sterility.[30] Furthermore, using ice or tissues to continue sex work during menstruation has been proven to be a source of bacterial infections.[31] These needs could be addressed in the community clinics of the two NGOs. However, as shown in other studies, FSWs faced many obstacles to accessing SRH care, due to the high costs or distance of the sites,[32] the stigmatising and discriminating attitudes of some health practitioners, the FSWs' social and economic marginalisation, and restrictive laws related to their activity.[33] FSWs thus preferred advice from their peers or self-medication. Moreover, peer educators from both NGOs expressed that public policies and international donors in Côte d'Ivoire currently focus on the identification and referral of new cases of HIV-positive FSWs, while HIV-negative women have limited access to care as stated above. A PrEP programme requires a medical follow-up every 3 months and thus implies to consider the regular follow-up of HIV-negative women. All efficacy PrEP trials provided a range of sexual healthcare services in addition to PrEP drugs. By design, these services were conditional to PrEP use. When transitioning to real life, such PrEP programmes reproduced such service model. Our results suggest that a paradigm shift towards a patient-centred approach should be preferred, that is, offering SRH services (such as contraception or STI testing and treatment) in which PrEP is an option but not mandatory. SRH services could also be a way to engage FSWs not ready for PrEP into regular care.

In addition, FSWs in San Pedro appeared to be in a more precarious situation than those in Abidjan due to their lower education level, higher number of children, irregularity of work, multiplicity of clients and work locations, and the lower price of sexual intercourse. They were also more likely to have condomless intercourse, notably for a large sum of money, and to report having had an STI over the past year. This can be explained by the fact that a large percentage of FSWs in San Pedro came to the area during the period of coffee and cocoa exploitation (September–December), which brought many migrant workers; this results in less stability and security. The high mobility of these women generates 'seasons of risk',[34] that is, times when an individual might face an increased risk of HIV infection. It is paramount to take this into account when implementing daily PrEP for these women[35]; they are the population most likely to regularly suspend their PrEP use. Developing mobile clinics that deliver HIV and SRH care services directly at prostitution sites could mitigate the issue of FSWs' mobility and address barriers to access to care, such as distance between prostitution sites and clinics or stigmatisation associated with their activity.

FSWs' needs for PrEP cannot be understood without additionally considering the broader contexts in which their risk of exposure to HIV is situated: the context of their work, their relationships and their concerns about family planning or stigmatisation, and so on. This is important information to consider if a PrEP programme is to successfully serve this at-risk population. In order to address women's experiences and concerns, a global SRH care package delivered through both community clinics and mobile clinics on prostitution sites appears essential. Several studies related to PrEP and SRH needs of FSWs showed as well that combination prevention approaches are necessary. First, as pointed by a study in Zimbabwe,[36] women need to perceive the risk of getting infected by HIV and to be able to access health services in order to take PrEP daily. Second, as shown by Dhana et al[37] in a systematic review, there is a lack of coordination between SRH and HIV services dedicated to FSWs in Africa when they are two distinct services; HIV services delivery models should integrate SRH services. Furthermore, in order to minimise stigma related to entry into care, services for HIV-positive and HIV-negative should not be dissociated.

A strength of this study is the use of mixed methods, allowing us to better describe and understand the challenges of PrEP implementation among FSWs in Côte d'Ivoire, as well as the strong collaboration with two NGOs helping us to reach the FSWs. Yet this study has some limitations. First, as it focused on FSWs reached by two NGOs, the included population was probably more likely to know about HIV prevention and to access SRH care. In addition, our sample did not include occasional or undeclared FSWs. As such, the results cannot be extrapolated to all FSWs working inside or around Abidjan and San Pedro but can provide an operational perspective for developing healthcare services. Peer educators conducted the survey during their on-site activities, which already consistently extended their working time. For logistic reasons, it was not possible to monitor the number of FSWs present on site, potentially eligible, examined for eligibility and included in the survey. Therefore, we are not able to provide participation rates. However, peer educators reported that most FSWs confirmed for eligibility did accept to answer the questionnaire. The survey was stopped when we reached the expected number of 1000 FSWs (600 in Abidjan and 400 in San Pedro) and all FSWs included in the survey were analysed. Finally, the use of FGDs could have led to socially desirable answers. A matter of concern, pointed out by fieldworkers and data collected, are the young underaged FSWs (three interviews were conducted with FSWs aged younger than 18 on the beaches of Abidjan). Ivorian law authorises HIV testing for teenagers aged 16 and 17 without parental consent. What about other care and services that cannot be delivered to them in the absence of consent?

## CONCLUSIONS

Implementing PrEP among FSWs in West Africa, such as in Côte d'Ivoire, is not only about providing a new prevention tool but is also an invitation to consider the regular follow-up of HIV-negative FSWs. A global care package should be offered to FSWs, including HIV prevention and care, STI screening and treatment, contraception,

menstrual management counselling and Hepatitis B Virus (HBV) screening, vaccination, and medical treatment. In addition, PrEP initiation should not limit access to SRH services; conversely, SRH services could be a way to attract FSWs to HIV prevention. Beyond reducing the risk of HIV among FSWs and their partners, PrEP provides an opportunity to improve their health condition more globally.

While current policies focus only on HIV-infected women and on the importance of testing new FSWs, our results highlight the importance of developing a people-focused approach, as opposed to an 'HIV-focused approach', that integrates all SRH needs when transitioning from PrEP efficacy trials to implementation.[19]

**Author affiliations**
[1]INED (French Institute for Demographic Studies), Paris, France
[2]CEPED, Centre for Population and Development, (Paris Descartes University, IRD, Inserm), Paris, France
[3]Programme PAC-CI/ANRS Research Site, Abidjan, Côte d'Ivoire
[4]ISPED, Inserm Research Center 1219 (Bordeaux Population Health), Université de Bordeaux, Bordeaux, France
[5]Espace Confiance, Abidjan, Côte d'Ivoire
[6]Aprosam, San Pedro, Côte d'Ivoire

**Acknowledgements** We would like to thank all participants as well as Aprosam's and Espace Confiance's peer educators.

**Collaborators** Members of the ANRS 12361 PrEP-CI study group: Aboubakar Sangaré (Aprosam, San Pedro, Côte d'Ivoire), Anglaret Xavier (PAC-CI, Abidjan, Côte d'Ivoire / Inserm, Bordeaux, France), Anoma Camille (Espace Confiance, Abidjan, Côte d'Ivoire), Barin Francis (Université François Rabelais, Tours, France), Bazin Brigitte (ANRS, Paris, France), Becquet Valentine (Ceped/IRD, Paris, France), Dabis François (ISPED/Inserm, Bordeaux, France), Danel Christine (PAC-CI, Abidjan, Côte d'Ivoire / Inserm, Bordeaux, France), Eholie Serge (PAC-CI, Abidjan, Côte d'Ivoire), Ekouevi Didier (PAC-CI, Abidjan, Côte d'Ivoire), Fonsart Julien (Hôpital Saint-Louis, Paris, France), Gbosi Kate (Aprosam, San Pedro, Côte d'Ivoire), Kwamé Abo (Programme National de Lutte contre le Sida, Côte d'Ivoire), Larmarange Joseph (Ceped/IRD, Paris, France), Masumbuko Jean-Marie (PAC-CI, Abidjan, Côte d'Ivoire), Méda Nicolas (Centre Muraz, Bobo-Dioulasso, Burkina Faso), Moh Raoul (PAC-CI, Abidjan, Côte d'Ivoire), Molina Jean-Michel (Hôpital Saint-Louis, Paris, France), N'dri-Yoman Thérèse (PAC-CI, Abidjan, Côte d'Ivoire), Nouaman Marcellin (PAC-CI, Abidjan, Côte d'Ivoire), Plazy Mélanie (ISPED / Inserm, Bordeaux, France), Soh Kouamé (Aprosam, San Pedro, Côte d'Ivoire), Tanoe Solange (Espace Confiance, Abidjan, Côte d'Ivoire), Yeo Roselyne (Espace Confiance, Abidjan, Côte d'Ivoire).

**Contributors** JL, SPE and CD designed the ANRS 12361 PrEP-CI study. J-MM and MN implemented the quantitative survey with the support of CA and SK. VB conducted the qualitative interviews. VB and JL developed the research question addressed in this paper. VB did the qualitative analysis, and MN did the statistical analysis. VB wrote the manuscript with the support of JL, MP and MN. All authors contributed to the interpretation and presentation of the findings. All authors approved the final version of the manuscript for submission.

**Funding** The PrEP-CI ANRS 12361 was funded by the Bill and Melinda Gates Foundation (Investment ID: OPP1106343) and the French National Agency for AIDS and Viral Hepatitis Research (ANRS).

**Competing interests** None declared.

**Patient consent for publication** Not required.

**Ethics approval** Research authorisations were obtained from the National Committee of Research Ethics within the Ivorian Ministry of Health and Public Hygiene (reference number: 057/MSHP/CNER-kp, delivered on 28 June 2016). Confidentiality was maintained and data were anonymised. Written informed consent was obtained by the investigator before each interview or before answering the questionnaire.

**Provenance and peer review** Not commissioned; externally peer reviewed.

**Data availability statement** Dataset are available upon request on Zenodo (DOI: 10.5281/zenodo.2269160).

**ORCID iD**
Valentine Becquet http://orcid.org/0000-0002-3599-5445

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
