## [Reviewer comments · BMJ Open]

ARTICLE DETAILS

TITLE (PROVISIONAL)	Sexual health needs of female sex workers in Côte d'Ivoire: a mixed-methods study to prepare the future implementation of pre-exposure prophylaxis (PrEP) for HIV prevention
AUTHORS	Becquet, Valentine; Nouaman, Marcellin; Plazy, Mélanie; Masumbuko, Jean-Marie; Anoma, Camille; Kouame, Soh; Danel, Christine; Eholie, Serge; Larmarange, Joseph

VERSION 1 – REVIEW

REVIEWER	Suzanne Day Postdoctoral Research Associate, Institute for Global Health and Infectious Diseases and the Department of Social Medicine, University of North Carolina at Chapel Hill, USA
REVIEW RETURNED	09-Jan-2019

GENERAL COMMENTS	Thank you for the opportunity to review the revised manuscript. I appreciate the efforts the authors have made to address concerns with the previous submission. In most instances these revisions have helped to strengthen the paper, which serves as a useful contribution to our understanding of the sexual and reproductive health needs of FSWs in Côte d'Ivoire – a crucial context to understand for the purpose of implementing future PrEP initiatives. There are however several points of clarification that I would recommend the authors consider in order to further strengthen the paper, particularly in the discussion section. Although heavily revised, the Discussion still requires some work in order to connect the analysis with the data presented in the Results section: 1. Page 4, Line 26-27: This sentence starts “Our research team was invited to explore the situation among FSWs in this country...” but this is quite vague. I would recommend specifying that you were invited to explore the sexual health needs of FSWs in this country, as otherwise the sentence reads as though a PrEP project has already started. This will help to more clearly explain the rationale behind the paper, which as I understand is to build our knowledge of FSWs’ experiences with sexual health in order to better inform the eventual roll-out of a PrEP program. Currently this rationale is not clearly stated, which makes the link between the current study and PrEP difficult to understand. Line 44 also uses vague language of describing the “additional needs” of FSWs, which should be changed to “sexual health care needs”.2. Page 5, Lines 7, 39 and 47: Should be ‘convenience sample’ (as in a sample produced using a convenience sampling technique).
--

	3. Page 8, Lines 56-57: This should be moved to the Discussion and elaborated upon (women’s difficulties negotiating condom use with partners). 4. Page 9, Lines 3-9: Would recommend reversing the order of the first two paragraphs (present the results on exposure to HIV before presenting the results on HIV testing). 5. Page 12, Line 23: The authors cite that in regards to FSWs not using condoms with intimate partners, these women “experience low decision-making power when facing the primacy of men’s sexual pleasure (17), in a context where gender norms reinforce male domination over women (18).” However, despite the edits made to this section these citations do not suffice as an explanation for why women have condomless sex with their partners. Based on the data presented, women were indeed coerced into condomless sex in order to ‘prove’ they trusted their partners, but women also used condomless sex as a negotiation strategy to obtain protection from their partners against the threat of violence. A clearer explanation is needed here that reflects the data actually presented in the results section. Currently the explanation of male sexual domination is too simple and does not capture the nuances of the women’s responses. 6. Page 12, Line 32: Citation #21 feels out of place: the discussion here is about negotiating condom use (or experiencing reduced capacity to do so), not about condom breakage. I would recommend either removing or further clarifying how this relates to the study findings. 7. Page 12, Lines 36-42: This discussion of HIV incidence rates does not fit with the previous discussion in this paragraph regarding women’s experiences with condom use/negotiation. This information needs to be reorganized into its own paragraph and presented in a way that more clearly links the study findings to the conclusion that PrEP could be an appropriate tool for use in this population. 8. Page 12, Lines 50-51: The authors state that they “were attentive to challenges that might hinder PrEP uptake and adherence for FSWs”; in addition to the need for medical follow-up, what other barriers to uptake/adherence were explored in this study? Would be useful to present greater information on this in the results; currently there is only a brief mention that "several questions emerged" for the women in interviews (Page 9, lines 13-15). 9. Page 13, Lines 37-49: The authors note that “Our results bring two considerations. First, in order to minimize stigma related to entry into care, services for HIV-positive and services for HIV-negative should not be dissociated.” It not clear that the issue of stigma was explored at all in the results of this study in relation to the participants’ interest in PrEP or need for SRH services. This paragraph needs to be more reflective of the study results as presented in the previous section. 10. Page 13, Lines 49-51: The following sentence needs work: “Second, rather than a PrEP program with additional services, a paradigm shift toward a patient-focused approach is needed, offering SRH services in which PrEP is an option but not
--	--

	mandatory.” It is not clear how this would represent a shift towards a “patient-focused approach”. This concept needs to be explained more clearly if it is to be used here. Additionally, it should be clarified that what the authors are suggesting, as I understand it, is the need for SRH services that are delivered without adherence to PrEP as a mandatory condition in order to obtain those services. More elaboration is needed as to how the current study results connect to this recommendation. 11. Page 14, Line 23: The conclusion asserts that “While current policies focus on only HIV-infected women and on the importance of testing new FSWs...” however, evidence to substantiate this point is not presented in the rationale/intro of the paper. I recommend elaborating on this point or revising the phrasing here to better reflect back on the rationale presented originally in the introduction.
--	--

REVIEWER	Janneke P. Bil Public Health Service of Amsterdam, the Netherlands My institute received restricted and unrestricted grants from Gilead Sciences, Inc. for studies that I have worked on within my institute.
REVIEW RETURNED	22-Jan-2019

GENERAL COMMENTS	I have read the article “Sexual health needs of female sex workers reached by two NGOs in Côte d’Ivoire: considerations for the future implementation of PrEP” with great interest. The strengths of the study are the number of people that filled in the survey and participated in the interviews and FGD and the results provide some insight in the sexual health needs of female sex workers. However, the paper needs some major revision on the following critical points before it can be accepted for publication: Abstract:  - The results of the study can be more clearly described. Some results seem contradictory (“clients use condoms with clients” and “some accepted condomless sexual intercourse for a large sum of money” and “inconsistent condom use....”). I would suggest writing something like this: “Although most FSW described consistent condom use with clients, some accepted condomless sexual intercourse for a large sum of money. Furthermore, condom use with their steady partner and knowledge of their partner’s HIV status was low”. Also some conclusions are presented as results (for example, FSW are highly exposed to HIV” and “FSWs faced many unmet needs regarding SRH”. Please move these conclusions to the results (see also comment on the result section). - The conclusions do not reflect the results presented in the abstract. I would revise this to for example: “Due to their inconsistent condom use FSW are highly exposed to HIV. Furthermore, FSW seem to face several barriers in accessing SRH. PrEP.....” Introduction:  - Please change MSMs for MSM (MSM is already plural). - The aim of the survey questionnaire and interviews/FGD could be more clearly described in the introduction. Especially the rationale for using a mixed-methods study is missing (Why did you choose a mixed methods study and how do they compliment each other?).
--

	Methods:  - Section “quantitative analysis of a survey questionnaire”, last sentence of the first paragraph: it is unclear what is measured with “HIV infection” - the description of the qualitative part of the study is insufficiently described. Please use the COREQ checklist for interviews or FGD (or another checklist) to describe the methods of the study sufficiently. For example the following elements are missing/unclear:  - How long did the interviews take? - What was the aim of the qualitative part of the study? - What is meant by a sociodemographer? - Was an interview guide used and which kind (semi-structured, structured etc)? - How many people analyzed the data and what is their background? Results:  - The themes of the qualitative study are interpretations and conclusions of the data analyses rather than they are the results of the analyses. The themes should reflect objective results and the conclusions and interpretation of the results should follow in the conclusion (see also comment on abstract). - the last sentence of the section “high HIV exposure despite the use of condom”: “...even if it required a medical follow-up every three months (99.4%)”. This sentence is confusing. Please indicate that the 99.4% of the people interested in PrEP would agree to a medical follow up every 3 months. - The results seem limited when compared to the items and interview guide described in the method section. It seems that not all results have been reported. Especially the data from the qualitative part seem to be very limited (considering the number of people included in this part of the study). Please give an explanation for this. Discussion:  - The description of the limitations of the study is limited. The use of FGD for example could have led to socially desirable answers. Also, the fact that PrEP was not more thoroughly explained and asked during the survey and/or interviews/FGD limits the results and the conclusion about PrEP implementation among FSW.
--	---

REVIEWER	Professor Hafiz Khan University of West London
REVIEW RETURNED	11-Jul-2019

GENERAL COMMENTS	I have particularly looked at statistics and data analysis section of the paper. My judgement is that data analyses using statistical methods are fine in the manuscript. I do not find any serious error in the paper.
---

VERSION 1 – AUTHOR RESPONSE

Reviewer(s)' Comments to Author:

Reviewer: 1

Reviewer Name: Suzanne Day

Institution and Country: Postdoctoral Research Associate, Institute for Global Health and Infectious Diseases and the Department of Social Medicine, University of North Carolina at Chapel Hill, USA
Please state any competing interests or state 'None declared': None declared.

Please leave your comments for the authors below

Thank you for the opportunity to review the revised manuscript. I appreciate the efforts the authors have made to address concerns with the previous submission. In most instances these revisions have helped to strengthen the paper, which serves as a useful contribution to our understanding of the sexual and reproductive health needs of FSWs in Côte d'Ivoire – a crucial context to understand for the purpose of implementing future PrEP initiatives. There are however several points of clarification that I would recommend the authors consider in order to further strengthen the paper, particularly in the discussion section. Although heavily revised, the Discussion still requires some work in order to connect the analysis with the data presented in the Results section:

1. Page 4, Line 26-27: This sentence starts “Our research team was invited to explore the situation among FSWs in this country...” but this is quite vague. I would recommend specifying that you were invited to explore the sexual health needs of FSWs in this country, as otherwise the sentence reads as though a PrEP project has already started. This will help to more clearly explain the rationale behind the paper, which as I understand is to build our knowledge of FSWs’ experiences with sexual health in order to better inform the eventual roll-out of a PrEP program. Currently this rationale is not clearly stated, which makes the link between the current study and PrEP difficult to understand. Line 44 also uses vague language of describing the “additional needs” of FSWs, which should be changed to “sexual health care needs”.

We corrected both sentences accordingly: “Our research team was invited to explore the sexual health needs of FSWs in this country as a potential target for a future PrEP program, knowing that PrEP was not yet available.” *and* “In this context, in order to design a future PrEP program targeting FSWs, the ANRS 12361 PrEP-CI pilot study was implemented to explore sexual health care needs that should be considered within such a program and to better describe FSWs currently reached by peer educators.”

2. Page 5, Lines 7, 39 and 47: Should be ‘convenience sample’ (as in a sample produced using a convenience sampling technique).

We corrected these mistakes.

3. Page 8, Lines 56-57: This should be moved to the Discussion and elaborated upon (women’s difficulties negotiating condom use with partners).

OK, this has been erased in the results section and elaborated upon in the discussion section: “First, the large majority did not use condoms with their regular partner despite their acknowledged concurrent sexual partnerships. Some women experienced coercion on the part of their male partners, questioning their faith in the relationship; having condomless sex was a proof of trust that was difficult to negotiate. Others used condomless sex as a negotiation strategy to obtain protection from their partners against the threat of violence. In a context where gender norms reinforce male domination over women (16), they consciously took risks when facing the primacy of men’s sexual pleasure (17).”

4. Page 9, Lines 3-9: Would recommend reversing the order of the first two paragraphs (present the results on exposure to HIV before presenting the results on HIV testing).

We reversed the order of the two paragraphs accordingly.

5. Page 12, Line 23: The authors cite that in regards to FSWs not using condoms with intimate partners, these women “experience low decision-making power when facing the primacy of men’s sexual pleasure (17), in a context where gender norms reinforce male domination over women (18).” However, despite the edits made to this section these citations do not suffice as an explanation for why women have condomless sex with their partners. Based on the data presented, women were indeed coerced into condomless sex in order to ‘prove’ they trusted their partners, but women also used condomless sex as a negotiation strategy to obtain protection from their partners against the threat of violence. A clearer explanation is needed here that reflects the data actually presented in the results section. Currently the explanation of male sexual domination is too simple and does not capture the nuances of the women’s responses.

This has been modified to better capture the women’s responses: “First, the large majority did not use condoms with their regular partner despite their acknowledged concurrent sexual partnerships. Some women experienced coercion on the part of their male partners, questioning their faith in the relationship; having condomless sex was a proof of trust that was difficult to negotiate. Others used condomless sex as a negotiation strategy to obtain protection from their partners against the threat of violence. In a context where gender norms reinforce male domination over women (16), they consciously took risks when facing the primacy of men’s sexual pleasure (17).”

6. Page 12, Line 32: Citation #21 feels out of place: the discussion here is about negotiating condom use (or experiencing reduced capacity to do so), not about condom breakage. I would recommend either removing or further clarifying how this relates to the study findings.

The sentence has been erased to simplify the message and highlight only key findings.

7. Page 12, Lines 36-42: This discussion of HIV incidence rates does not fit with the previous discussion in this paragraph regarding women’s experiences with condom use/negotiation. This information needs to be reorganized into its own paragraph and presented in a way that more clearly links the study findings to the conclusion that PrEP could be an appropriate tool for use in this population.

We wrote a new paragraph to better link the study findings and our conclusion that PrEP could be an appropriate tool: “Despite the adoption of preventive behavior (condom use) in most cases, FSWs are still highly exposed to HIV, due their high number of sexual partners and the occurrence of remaining unprotected sexual acts. In a complementary study within the PrEP-CI project (not yet published), we estimated the incidence among the 1000 surveyed FSWs using a recent infection testing algorithm adapted to the Ivorian context: we found an incidence of 2.2 per 100 person-years (1.5 in Abidjan and 3.2 in San Pedro) (21). In such context, oral PrEP could be an appropriate and complementary preventive tool to cover the situations where condom cannot be negotiated”

8. Page 12, Lines 50-51: The authors state that they “were attentive to challenges that might hinder PrEP uptake and adherence for FSWs”; in addition to the need for medical follow-up, what other barriers to uptake/adherence were explored in this study? Would be useful to present greater information on this in the results; currently there is only a brief mention that “several questions emerged” for the women in interviews (Page 9, lines 13-15).

In the results, there is a short description of such questions. We added this in the discussion: “For this reason we were attentive to challenges that might hinder PrEP uptake and adherence for FSWs, such as side-effects or timing of daily administration.” *We don’t have enough space here to further describe these barriers, which are only hypothetical anyway as PrEP has not been implemented yet.*

9. Page 13, Lines 37-49: The authors note that “Our results bring two considerations. First, in order

to minimize stigma related to entry into care, services for HIV-positive and services for HIV-negative should not be dissociated.” It not clear that the issue of stigma was explored at all in the results of this study in relation to the participants’ interest in PrEP or need for SRH services. This paragraph needs to be more reflective of the study results as presented in the previous section.

Stigma was explored at the end of the last part of the results and in several parts of the discussion. We clarified the results (end of : “Beyond HIV, many unmet SRH needs exist” section): “In interviews, some FSWs reported the inconvenient opening times and/or location, the fear of being identified as an FSW in the clinic area and the stigmatizing and judgmental attitudes of health professionals as reasons for not visiting these clinics.”

10. Page 13, Lines 49-51: The following sentence needs work: “Second, rather than a PrEP program with additional services, a paradigm shift toward a patient-focused approach is needed, offering SRH services in which PrEP is an option but not mandatory.” It is not clear how this would represent a shift towards a “patient-focused approach”. This concept needs to be explained more clearly if it is to be used here. Additionally, it should be clarified that what the authors are suggesting, as I understand it, is the need for SRH services that are delivered without adherence to PrEP as a mandatory condition in order to obtain those services. More elaboration is needed is as to how the current study results connect to this recommendation.

This part has been moved earlier in the discussion and entirely rephrased: “All efficacy PrEP trials provided a range of sexual healthcare services in addition to PrEP drugs. By design, these services were conditional to PrEP use. When transitioning to real life, such PrEP programs reproduced such service model. Our results suggest that a paradigm shift toward a patient-centered approach should be preferred, that is offering sexual and reproductive health services (such as contraception or STI testing and treatment) in which PrEP is an option but not mandatory. SRH services could also be a way to engage FSWs not ready for PrEP into regular care.”

11. Page 14, Line 23: The conclusion asserts that “While current policies focus on only HIV-infected women and on the importance of testing new FSWs...” however, evidence to substantiate this point is not presented in the rationale/intro of the paper. I recommend elaborating on this point or revising the phrasing here to better reflect back on the rationale presented originally in the introduction.

We added two sentences (and two references) at the beginning of the introduction, to substantiate this point in the intro of the paper:

“Despite global progress in reducing new HIV infections and AIDS-related deaths in the last 10 years in sub-Saharan Africa, current policies and programmes are focusing on the identification of HIV-infected people in order to link them to HIV care and treatment, knowing that antiretroviral treatment has been proven to reduce HIV transmission (1,2). However, the number of new HIV infections still remain too high to achieve epidemic control.”

Reviewer: 2

Reviewer Name: Janneke P. Bil

Institution and Country: Public Health Service of Amsterdam, the Netherlands

Please state any competing interests or state ‘None declared’: My institute received restricted and unrestricted grants from Gilead Sciences, Inc. for studies that I have worked on within my institute.

Please leave your comments for the authors below

I have read the article “Sexual health needs of female sex workers reached by two NGOs in Côte d’Ivoire: considerations for the future implementation of PrEP” with great interest. The strengths of the study are the number of people that filled in the survey and participated in the interviews and FGD

and the results provide some insight in the sexual health needs of female sex workers. However, the paper needs some major revision on the following critical points before it can be accepted for publication:

Abstract:

- The results of the study can be more clearly described. Some results seem contradictory (“clients use condoms with clients” and “some accepted condomless sexual intercourse for a large sum of money” and “inconsistent condom use...”). I would suggest writing something like this: “Although most FSW described consistent condom use with clients, some accepted condomless sexual intercourse for a large sum of money. Furthermore, condom use with their steady partner and knowledge of their partner’s HIV status was low”. Also some conclusions are presented as results (for example, FSW are highly exposed to HIV” and “FSWs faced many unmet needs regarding SRH”. Please move these conclusions to the results (see also comment on the result section).
- The conclusions do not reflect the results presented in the abstract. I would revise this to for example: | “Due to their inconsistent condom use FSW are highly exposed to HIV. Furthermore, FSW seem to face several barriers in accessing SRH. PrEP.....”

We revised the abstract accordingly.

Introduction:

- Please change MSMs for MSM (MSM is already plural).

We corrected this mistake.

- The aim of the survey questionnaire and interviews/FGD could be more clearly described in the introduction. Especially the rationale for using a mixed-methods study is missing (Why did you choose a mixed methods study and how do they complement each other?).

OK, we added the following sentences: “This paper aims to describe the work and social environment of FSWs, their SRH needs and possible barriers for accessing care in two different settings in Côte d’Ivoire; i.e. different elements that need to be taken into account when implementing PrEP. We adopted a mixed approach: a quantitative survey was used to reach a high number of FSWs in order to be able to calculate incidence of HIV infection and to compare sexual and health behaviors and needs of FSWs in the two settings; qualitative interviews were conducted to understand rationales behind these behaviors and needs.”

Methods:

- Section “quantitative analysis of a survey questionnaire”, last sentence of the first paragraph: it is unclear what is measured with “HIV infection”

It was rephrased: “HIV infection” has been replaced by “HIV monitoring”, and we developed the meaning into the brackets “(regularity of HIV testing, knowledge of HIV status of the regular partners)”.

- the description of the qualitative part of the study is insufficiently described. Please use the COREQ checklist for interviews or FGD (or another checklist) to describe the methods of the study sufficiently. For example the following elements are missing/unclear:

- How long did the interviews take?

This has been added “Data were collected at prostitution sites in and around Abidjan and San Pedro, by a female researcher in demography (first author) who carried out 22 in-depth interviews (duration:

30 to 60 minutes) and eight focus group discussions (FGDs) (duration: 60 to 150 minutes) using a semi-structured interview guide.”

- What was the aim of the qualitative part of the study?

This has been added in the introduction.

- What is meant by a sociodemographer?

Sociodemography is a division of Demography that studies societal causes of population dynamics. We replaced the word by “female researcher in Demography” to simplify the meaning, as it might be a French term that has no translation in English studies. Gender was added as expected in the COREQ checklist.

- Was an interview guide used and which kind (semi-structured, structured etc)?

It has been added in the following sentence: “Data were collected at prostitution sites in and around Abidjan and San Pedro, by a female researcher in demography (first author) who carried out 22 in-depth interviews (duration: 30 to 60 minutes) and eight focus group discussions (FGDs) (duration: 60 to 150 minutes) using a semi-structured interview guide.”

- How many people analyzed the data and what is their background?

It has been added: “Each in-depth interview and FGD was recorded (except for two participants who refused it as they were afraid to be recognized), transcribed and uploaded into NVivo software (QSR International Pty Ltd. Version 11 Pro, 2016) by the qualitative interviewer. She also conducted the qualitative analysis following two principles.”

Results:

- The themes of the qualitative study are interpretations and conclusions of the data analyses rather than they are the results of the analyses. The themes should reflect objective results and the conclusions and interpretation of the results should follow in the conclusion (see also comment on abstract).

We erased one sentence to include it in the discussion section: “It seemed that the women were not in a strong negotiation position, and accepted to have condomless sex despite knowing their partners had concomitant relationships”. Other conclusions “high exposure to HIV”, “many unmet needs” are titles of subsections but not data analyses.

- the last sentence of the section “high HIV exposure despite the use of condom”: “...even if it required a medical follow-up every three months (99.4%)”. This sentence is confusing. Please indicate that the 99.4% of the people interested in PrEP would agree to a medical follow up every 3 months.

We modified the sentence accordingly.

- The results seem limited when compared to the items and interview guide described in the method section. It seems that not all results have been reported. Especially the data from the qualitative part seem to be very limited (considering the number of people included in this part of the

study). Please give an explanation for this.

Indeed, reporting mixed-methods results limit the possibility to develop fully qualitative results. They will be included in another paper. We added an explanation in the qualitative method section: "Data collected provide great information on FSWs' sexual and health behaviors and needs, preferences and social trajectories. However, many themes are not included in this paper and will be addressed in a further paper."

Discussion:

- The description of the limitations of the study is limited. The use of FGD for example could have led to social desirable answers. Also, the fact that PrEP was not more thoroughly explained and asked during the survey and/or interviews/FGD limits the results and the conclusion about PrEP implementation among FSW.

The sentence on the use of FGDs has been added.

PrEP has been thoroughly explained and asked during the qualitative interviews/FGDs though; we clarified this point in the results section. "During each qualitative interview and FGD, we presented PrEP as a medicine that could protect them against HIV if properly taken (...)"

Reviewer: 3

Reviewer Name: Professor Hafiz Khan

Institution and Country: University of West London

Please state any competing interests or state 'None declared': None

Please leave your comments for the authors below

I have particularly looked at statistics and data analysis section of the paper. My judgement is that data analyses using statistical methods are fine in the manuscript. I do not find any serious error in the paper.

VERSION 2 – REVIEW

REVIEWER	Suzanne Day University of North Carolina at Chapel Hill, USA
REVIEW RETURNED	25-Nov-2019

GENERAL COMMENTS	Thank you for the opportunity to review this revised version of the manuscript. I appreciate the authors' efforts to address the concerns noted on the previous version. There are a few spots where additional details/corrections would be helpful, and I do still feel that there is an issue with how some of the results are characterized in the Discussion section (though the authors have improved it substantially). My specific comments are as follows: Abstract: when noting that the prevalence of contraceptive use was low, it should be specified that this refers to contraceptive strategies other than condoms. Introduction, line 18: "recommended by the World Health Organisation (WHO) for population at" - should be "populations at".
--

	Page 6, lines 40-41: When describing survey eligibility criteria, would change "never being diagnosed HIV-positive" to more accurately specify what the inclusion criteria were: did women have to currently be known as HIV negative to participate (i.e. via recent test)? Or could women whose serostatus was unknown also participate? If the later, would rephrase to: "and who were HIV negative or of unknown HIV status at the time of the survey." Page 7, lines 37-39: Noting the composition of interviews and focus groups, "Every time, according to the type of interviews to be performed, one to three FSWS agreed to be interviewed individually or five to eight FSWs agreed to be interviewed as a group." This is confusing; does this mean that interview groups consisted of between 1-3 participants at a time, while focus groups consisted of 5-8 participants? (Also, careful to avoid capitalizing the second "S" in FSWs.) Page 11, line 5: "So, if condom use was high in general" - the information presented above notes that condom use was not "high in general", but rather context-dependent. Page 11, line 18: "and implying to have a regular medical follow-up." Not sure what this means. Would suggest rephrasing to "we presented PrEP as a medicine that could protect against HIV if properly taken, and explained that it would require regular medical follow-up" if this is indeed what was explained to participants. Discussion, line 37: I appreciate the additional discussion of the results regarding women's use of condomless sex to navigate the difficult issue of establishing trust with their partners and avoiding the threat of violence. However, I still do not think that it is accurate to explain your participants' behavior this way: "In a context where gender norms reinforce male domination over women (20), they consciously took risks when facing the primacy of men's sexual pleasure (21)." According to the evidence presented in the results, it is not "the primacy of men's sexual pleasure" that women were facing, it was specifically the threat of violence. Indeed it seemed to be a calculated risk-mitigation strategy (i.e. the risk of violence), though of course this then exposes women to the risk of HIV transmission. I would recommend removing this line as it undermines the previous discussion of the nuances involved in negotiating trust and mitigating threats of violence. Page 14 (Discussion), line 60: "FSWs are still highly exposed to HIV, due their high number of sexual partners" - their risk of HIV was due to the occurrence of remaining unprotected sex, not a high # of partners. Number of partners is not substantiated as a risk activity in any of the evidence presented in the Results. Page 15, line 9-10: "situations where condom cannot be negotiated" - should be "situations where condom use cannot be negotiated".
--	---

REVIEWER	Janneke P. Bil Public Health Service of Amsterdam, the Netherlands. My institute received restricted and unrestricted grants from Gilead Sciences, Inc. for studies that I have worked on within my institute.
-----------------	---

REVIEW RETURNED	30-Oct-2019
GENERAL COMMENTS	The authors have addressed the previously raised concerns sufficiently enough.

VERSION 2 – AUTHOR RESPONSE

Reviewer(s)' Comments to Author:

Reviewer: 2

Reviewer Name: Janneke P. Bil

Institution and Country: Public Health Service of Amsterdam, the Netherlands.

Please state any competing interests or state 'None declared': My institute received restricted and unrestricted grants from Gilead Sciences, Inc. for studies that I have worked on within my institute.

Please leave your comments for the authors below

The authors have addressed the previously raised concerns sufficiently enough.

Reviewer: 1

Reviewer Name: Suzanne Day

Institution and Country: University of North Carolina at Chapel Hill, USA

Please state any competing interests or state 'None declared': None declared.

Please leave your comments for the authors below

Thank you for the opportunity to review this revised version of the manuscript. I appreciate the authors' efforts to address the concerns noted on the previous version. There are a few spots where additional details/corrections would be helpful, and I do still feel that there is an issue with how some of the results are characterized in the Discussion section (though the authors have improved it substantially). My specific comments are as follows:

Thank you for reviewing the revised version of the manuscript, and for your comments that improved highly the quality of our paper. We answered your suggestions directly in your text.

Abstract: when noting that the prevalence of contraceptive use was low, it should be specified that this refers to contraceptive strategies other than condoms.

Indeed, it was unclear; we added this specification in the abstract.

Introduction, line 18: "recommended by the World Health Organisation (WHO) for population at" - should be "populations at".

We corrected the word accordingly.

Page 6, lines 40-41: When describing survey eligibility criteria, would change "never being diagnosed HIV-positive" to more accurately specify what the inclusion criteria were: did women have to currently be known as HIV negative to participate (i.e. via recent test)? Or could women whose serostatus was unknown also participate? If the later, would rephrase to: "and who were HIV negative or of unknown HIV status at the time of the survey."

Criteria were to be known as HIV negative or to be of unknown HIV status at the time of the survey. We specified this in the section related to the quantitative questionnaire, as suggested.

Page 7, lines 37-39: Noting the composition of interviews and focus groups, "Every time, according to the type of interviews to be performed, one to three FSWs agreed to be interviewed individually or five to eight FSWs agreed to be interviewed as a group." This is confusing; does this mean that interview groups consisted of between 1-3 participants at a time, while focus groups consisted of 5-8 participants? (Also, careful to avoid capitalizing the second "S" in FSWs.)

We meant that we performed 1 to 3 individual interviews with 1 FSW each time + 1 focus group with 5 to 8 FSWs. We rephrased that part: *"Every time, according to the type of interviews to be performed, we conducted one to three individual interviews and/or one focus group with five to eight FSWs."* We hope it is less confusing this way.

We also corrected the capital letter in "FSWs" and verified the whole text to make sure the mistake does not appear elsewhere.

Page 11, line 5: "So, if condom use was high in general" - the information presented above notes that condom use was not "high in general", but rather context-dependent.

The sentence was inaccurate so we rephrased it: *"Even if the majority of FSWs declared regular condom use, most of them were still exposed to HIV"*

Page 11, line 18: "and implying to have a regular medical follow-up." Not sure what this means. Would suggest rephrasing to "we presented PrEP as a medicine that could protect against HIV if properly taken, and explained that it would require regular medical follow-up" if this is indeed what was explained to participants.

We rephrased that sentence accordingly, as it is indeed what was explained to participants.

Discussion, line 37: I appreciate the additional discussion of the results regarding women's use of condomless sex to navigate the difficult issue of establishing trust with their partners and avoiding the threat of violence. However, I still do not think that it is accurate to explain your participants' behavior this way: "In a context where gender norms reinforce male domination over women (20), they consciously took risks when facing the primacy of men's sexual pleasure (21)." According to the evidence presented in the results, it is not "the primacy of men's sexual pleasure" that women were facing, it was specifically the threat of violence. Indeed it seemed to be a calculated risk-mitigation strategy (i.e. the risk of violence), though of course this then exposes women to the risk of HIV transmission. I would recommend removing this line as it undermines the previous discussion of the nuances involved in negotiating trust and mitigating threats of violence.

We completely agree that FSWs were negotiating trust and/or protection with their partners, to mitigate threats of violence. The appointed sentence was not at the right place. However, we feel that it is still true and can explain why clients want to negotiate higher prices for condomless intercourses: they feel it would increase their sexual pleasure. For this reason, we moved the sentence a bit further in the discussion. We also added a short sentence related to the risk-mitigation strategy: *"First, the large majority did not use condoms with their regular partner despite their acknowledged concurrent sexual partnerships. Some women experienced coercion on the part of their male partners, questioning their faith in the relationship; having condomless sex was a proof of trust that was difficult to negotiate. Others used condomless sex as a negotiation strategy to obtain protection from their partners against the threat of violence. It seemed to be a calculated risk-mitigation strategy, although women were then exposed to the risk of HIV transmission. Second, some FSWs accepted*

condomless sexual intercourse for a large sum of money, especially when they had had few previous clients. Financial need associated with low prices of sexual intercourses and irregular weekly earnings drove some FSWs to engage in condomless sex as a way to earn more. In a context where gender norms reinforce male domination over women (20), they consciously took risks when facing the primacy of men's sexual pleasure (21). »

Page 14 (Discussion), line 60: "FSWs are still highly exposed to HIV, due their high number of sexual partners" - their risk of HIV was due to the occurrence of remaining unprotected sex, not a high # of partners. Number of partners is not substantiated as a risk activity in any of the evidence presented in the Results.

We deleted that part of the sentence, as it is not presented in the results indeed.

Page 15, line 9-10: "situations where condom cannot be negotiated" - should be "situations where condom use cannot be negotiated".

We rephrased that sentence accordingly.

VERSION 3 – REVIEW

REVIEWER	Suzanne Day Institute for Global Health and Infectious Diseases, University of North Carolina at Chapel Hill, USA.
REVIEW RETURNED	06-Dec-2019
GENERAL COMMENTS	I believe the authors have sufficiently addressed my prior concerns with their most recent revisions to the manuscript.